# Substance Use and Attendance Motives of Electronic Dance Music (EDM) Event Attendees: A Survey Study

**DOI:** 10.3390/ijerph20031821

**Published:** 2023-01-19

**Authors:** Edith Van Dyck, Koen Ponnet, Tina Van Havere, Bert Hauspie, Nicky Dirkx, Jochen Schrooten, Jon Waldron, Meryem Grabski, Tom P. Freeman, Helen Valerie Curran, Jan De Neve

**Affiliations:** 1Institute for Psychoacoustics and Electronic Music (IPEM), Department of Art History, Musicology and Theatre Studies, Ghent University, 9000 Ghent, Belgium; 2Research Group for Media, Innovation and Communication Technologies, Department of Communication Sciences, IMEC-MICT, Ghent University, 9000 Ghent, Belgium; 3Substance Use and Psychosocial Risk Behaviours (SUPR-B), University of Applied Sciences and Arts, 9000 Ghent, Belgium; 4VAD, 1030 Brussels, Belgium; 5Clinical Psychopharmacology Unit, University College London, London WC1E 7HB, UK; 6Addiction and Mental Health Group (AIM), Department of Psychology, University of Bath, Bath BA2 7AY, UK; 7Department of Data-Analysis, Ghent University, 9000 Ghent, Belgium

**Keywords:** electronic dance music, EDM, nightlife, substance use, drugs, attendance motives, motivation

## Abstract

EDM event attendees are a high-risk population for substance use and associated adverse effects. The aim of this study was to examine substance use at EDM events, focusing on associations between attendance motives and substance use. Sociodemographic characteristics, event specifics, past-year use, and attendance motives were assessed through an online survey. Participants were 1345 Belgian EDM event attendees (69.44% male, M_age_ = 22.63, SD_age_ = 4.03). Ecstasy/MDMA/Molly (52.28%), other synthetic hallucinogens (53.68%), ketamine (42.13%), amphetamines (40.45%), and alkyl nitrites (poppers) (32.76%) were most frequently used at festivals/outdoor parties/raves. In nightclubs, cocaine (32.29%) was shown to be prevalent as well, while other synthetic hallucinogens (15.79%) were less often consumed. At events with a more private character, cannabis (68.88%) and magic mushrooms (66.44%) were most frequently used. Aside from alcohol (47.76%), substance use in pubs/bars was negligible. Overall enjoyment was demonstrated to be the key attendance motive, which was succeeded by those relating to music and socialization. A wide range of motives proved to be more important to users (e.g., dance, exploration, escapism, excitement, alcohol, drugs) than non-users, while some were associated with the use of particular substances. The prevalence of substance use was shown to be dependent on the specifics of the setting. Moreover, the idea of a three-dimensional classification of the most principal motives for music event attendance was supported. Finally, correlations were estimated between attendance motives and substance use as well as specific substances. Results could enable more tailored approaches in prevention and harm reduction efforts as well as event management strategies.

## 1. Introduction

Spurred by technological developments supporting electronically engineered music since the 1980s, the popularity of electronic dance music (EDM) events has known an exponential growth all around the globe [1]. Originally, EDM was mostly played at raves, i.e., all-night dance parties, typically held in underground venues not intended for that specific purpose [2]. Today, large EDM festivals (e.g., Tomorrowland, Ultra Miami Music Festival) are among the most common types of music festivals in the still expanding festival industry [3,4,5]. EDM audiences are generally considered to be part of a rather distinct musical culture and overall, substance use is quite common at EDM gatherings [3,6,7]. At such events, party drugs (or ‘club drugs’) traditionally refer to illicit stimulant and hallucinogenic drugs commonly used to enhance the experience, including ecstasy/MDMA/Molly, powder cocaine, LSD, and ketamine [8,9,10], next to an ever-evolving range of synthetic psychedelic and stimulant “designer” drugs [11,12,13,14], and research indicates that drug prevalence at EDM events is still rising [9,15,16,17]. Correspondingly, EDM event audiences have proven to constitute a high-risk population for non-medical drug use [1,18,19,20,21,22,23], overdoses (i.e., of MDHMA, GHB, and other drugs, see e.g., [24,25]), alcohol overuse [26], as well as associated adverse effects such as driving accidents [19] and mass-casualty incidents [27].

Even though these events are widespread, target large audiences, and as such entail a variety of risks related to substance use, audience analysis research exploring EDM gatherings is yet sparse. However, event managers continuously strive to better understand the motives of attendees in order to design more fitting products and services and because they are a precursor of satisfaction and a factor in overall decision making [28]. In addition, a more profound understanding of visitors’ reasons to attend such events may provide stakeholders with a better grasp of their lines of thought and as such support substance-use related prevention and harm reduction efforts.

In psychology and sociology, the definition of motivation is directed toward emotional and cognitive motives [29] or internal and external ones [30]. Although it is only one of many variables that explain behavior (next to learning, cultural conditioning, social influences, and perceptions), motivation is regarded as “the impelling and compelling driving force behind all behavior” [31] and can thus be considered as the starting point that launches the decision process [32]. To explore the motivation process, motives—defined as internal factors that arouse, direct, and integrate an individual’s behavior [33,34,35]—were proposed as the factors through which researchers are able to establish theories to categorize and interpret various types of motivation processes [36]. Three different paradigms of understanding motivation through motive exploration—in previous research mostly applied to leisure activities such as tourism and event attendance—have been proposed, of which the most dominant one is the push–pull theory [37]. This theory considers a push motivation to derive from the discrepancy between the ideal state and the current one, thus, in other words, a disequilibrium within the individual (e.g., escape from the current environment, facilitation of social interaction). A pull motivation, on the other hand, is believed to stem from the inherent attractiveness of an activity (e.g., a particular artist at a music event) [31,33,38]. Interrelated with this theory is the escape-seeking dichotomy [39,40], which considers two motivational forces: seeking and escaping. Escaping refers to the desire to leave the everyday environment behind oneself, while seeking consists of the urge to obtain psychological (intrinsic) rewards through presence in a contrasting (new or old) environment. Although these dimensions are quite similar to the push (escape) and pull (seeking) dimensions, the escape–seeking dichotomy provides a refinement of the push–pull theory in terms of intrinsic benefits incorporated in the seeking force, while the concept of pull rather relates to attractions than to social–psychological needs. Thus, both personal (e.g., escaping personal troubles, seeking relaxation) and interpersonal dimensions (e.g., escaping co-workers, seeking varied and increased social interaction) are key here, and potential psychological benefits emanate from the interplay of both forces. Next to these paradigms, a third one was proposed by Getz [41,42], who classified the basic needs met by leisure activities into three categories: physical, interpersonal/social, and personal. It consists of an adapted version of Maslow’s hierarchy, which organized overall human needs into five categories—(in descending order of importance) physiological needs, safety, social, esteem, and self-actualization needs—and suggested that the appearance of one need usually depends on the satisfaction of a more fundamental need [43,44].

In line, motives to attend to leisure activities are usually multiple (e.g., [32,35,45]). Thus, rather than satisfying one specific desire, the decision to take part in an activity or attend a certain event can be commonly regarded as a directed action triggered by an aspiration to meet a number of needs. Research on the attendance motives of music events, which mostly focused on music festivals [46], indicates that people are indeed attracted to music events for several reasons [47]. Faulkner et al. [48], for example, reported three key motives for attendance: event excitement, novelty, and socialization. Somewhat in line, Bowen and Daniels [47] also identified three dimensions of attendance motivation, being: overall enjoyment, music, and socialization. Additionally, the engagement with a festival atmosphere and participation in diverse festival activities and workshops were indicated as main attendance motivators by Nicholson and Pearce [49]. In a study regarding two different music festivals in the UK, listening to music and watching an artist were exposed as principal motives for visiting one of the festivals, while the atmosphere and opportunities for socialization were key for attending the other [50]. Furthermore, focusing on an ethno music festival in Serbia, Blešić et al. [51] exposed four dominant reasons for attending music festivals, namely, festival perception and learning, exploration of festival program and atmosphere, psychophysical welfare (e.g., escape from routine), and socialization. A last exemplary investigation regarded Chinese music festivals and denoted items such as togetherness, musical enjoyment, and event novelty as pivotal attendance motives [52]. Overall, key motives of music event attendance gravitate toward socialization, musical enjoyment, and overall enjoyment/engagement with the atmosphere [4,47].

Due to the nature of the previously studied events, however, it remains unclear whether the above-described findings also apply to EDM audiences. Furthermore, taking the prevalence of substance use at such events into account, it could be questioned how observed motives relate to consumption patterns. In addition, as previous work on drug prevalence mostly focused on one particular EDM festival or club, hardly anything is known about the role of the setting. In the light of the above, the aim of this study is to explore the currently still expanding setting of the EDM event scene and related substance use. The study focuses on profiles of frequent attendees recruited both online and at a wide range of Belgian EDM events with respect to demographics, substance use, and nightlife engagement. The use of specific substances is tested and compared over different settings. Moreover, respondents’ motives to visit these events are explored and, finally, associations between these motives and substance use (users versus non-users, use of particular substances) are tested.

## 2. Materials and Methods

### 2.1. Data Collection and Inclusion Criteria

Using an online survey on substance use and nightlife engagement, data were gathered from May to November 2017. Respondents were recruited both offline and online, and eligibility criteria were: 18–34 years old; having attended at least six EDM events in the past 12 months; and residing in Belgium. The age range was determined to match the upper age limit of the European Monitoring Centre for Drug and Drug Addiction’s (EMCDDA) definition of a ‘young adult’ [53]. The number of events was chosen to ensure sufficient engagement with the nightlife scene [54].

Online recruitment was carried out primarily through paid, targeted advertisements on social media platforms (i.e., Facebook and Instagram). Ad viewing was enabled for users meeting the eligibility criteria and showing interest in or interacting with content related to the EDM scene (e.g., DJs, news groups, festivals, events, nightclubs). In addition, the survey was promoted via online groups, fora and websites focusing on EDM. Offline recruitment took place in popular nightclubs and at festivals situated in the EDM scene in Belgium. Clubs were selected using Resident Advisor [55], which is a widely used website encompassing reviews, news and ticket sales for EDM events. Since this website does not include festivals, nightlife experts were consulted to draft a list of key festivals in the EDM scene. At the selected nightclubs and festivals (who agreed to grant access for the study), participants were selected using a random intercept method (see [56]) in order to reduce the risk of selection bias. Fieldworkers were assigned fixed (unmarked) areas, large enough to ensure a regular flow of potential participants (i.e., approximately two by four meters in crowded areas; larger in less dense areas), and invited every second individual who entered this space. Visibly intoxicated individuals were not included. After filling out a screener to check for eligibility, potential respondents received the link to the online survey. In total, 1345 valid survey responses were collected. 

### 2.2. Measures

Demographic characteristics (age; gender; relationship status; country of residence; education; occupation), frequency of past-year EDM event attendance (0 = not in the last 12 months; 1 = 3 times or less; 2 = every two or three months; 3 = monthly; 4 = fortnightly; 5 = weekly; 6 = 3 times a week or more), and type of EDM events visited in the last year (nightclub; licensed festival/outdoor party/rave; unlicensed festival/outdoor party/rave; pub/bar; house party) were queried. Frequency of past-year use (including 20 substances/substance classes, alcohol, and the option to indicate other substances) was assessed (0 = not in the last 12 months; 1 = 3 times or less; 2 = every two or three months; 3 = monthly; 4 = fortnightly; 5 = weekly; 6 = 3 times a week or more), along with where these substances were mostly consumed (nightclub; licensed festival/outdoor party/rave; unlicensed festival/outdoor party/rave; pub/bar; house party; own/friend’s home; outside/public space). The escape–seeking dichotomy, push–pull theory, and Getz’s adapted version of Maslow’s hierarchy were employed to identify motives for EDM event attendance. The following motives were rated on a four-point Likert scale (0 = not important; 1 = not very important; 2 = slightly important; 3 = very important): ‘to dance’(Dance); ‘my friends are going’ (Friends); ‘to meet new people’ (Meet); ‘to look for sex’ (Sex); ‘to look for a partner’ (Partner); ‘to escape my daily life’ (Escape); ‘to take drugs’ (Drugs); ‘to drink alcohol’ (Alcohol); ‘to have fun’ (Fun); ‘to listen to music’ (Music); ‘to explore my mind’ (Explore); ‘to seek excitement’ (Excite); ‘to cope with my problems’ (Cope); ‘to open up to my friends’ (Open); ‘to see a particular artist’ (Artist).

### 2.3. Statistical Analyses

Data were analyzed using SPSS version 26 and R version 3.5.2 [57]. Spearman correlations were computed to account for the ordinal nature of the Likert-type data, while Wald-type confidence intervals were constructed for the proportions. Random intercept models were conducted to account for the multiple responses per participant.

## 3. Results

### 3.1. Demographics

The majority of the participants (N = 1345) were male (69.44%) and single (54.42%). The mean age was 22.63 years (SD = 4.03). Most participants were students (63.35%), while 26.54% were employed on a full-time and 6.84% on a part-time basis. Regarding education, 80.00% had completed or were attending higher education (Bachelor/Master). Participants attended a mean of 20.21 (SD = 19.69) EDM events in the past year. Overall, 96.05% of them indicated to have visited at least one licensed festival/outdoor party/rave, 90.18% at least one nightclub, 92.71% at least one pub/bar, 85.35% at least one house party, and 28.18% at least one unlicensed festival/outdoor party/rave.

### 3.2. Substance Use

Of all participants, 96.43% used alcohol and 64.24% drugs in the past year. The percentages of past-year substance use among participants are shown in Figure 1, revealing the most frequent use of alcohol, followed by cannabis, ecstasy/MDMA/Molly, and cocaine.

Table 1 displays Spearman correlations between consumption frequencies of the most prevalent substances (i.e., consumed by at least 5% of the participants in the past year). Significant correlations were retrieved between ecstasy/MDMA/Molly and all other substances, and between cannabis and a wide range of other substances (excluding alkyl nitrates (poppers), amphetamines, and alcohol).

Location of past-year use of the most popular substances is displayed in Table 2, demonstrating clear variations in distributions between different substances: magic mushrooms and cannabis were mostly used in domestic environments; alcohol predominantly in pubs/bars; amphetamines, ecstasy/MDMA/Molly, and ketamine mainly in nightclubs as well as at licensed festivals/outdoor parties/raves; other synthetic hallucinogens were primarily consumed at licensed festivals/outdoor parties/raves; cocaine mainly in nightclubs and domestic environments; LSD and nitrous oxide were mostly used at licensed festivals/outdoor parties/raves and at home; alkyl nitrites (poppers) were mainly taken at licensed festivals/outdoor parties/raves, in nightclubs, as well as at home.

### 3.3. Attendance Motives

As shown in Table 3, ‘to have fun’ was indicated to be the primary reason to attend EDM events, followed by (in descending order of importance): ‘to listen to music’, ‘to see a particular artist’, ‘to dance’, ‘my friends are going’, ‘to explore my mind’, ‘to escape my daily life’, and ‘to seek excitement’. Other motives proved to be less important, with ‘to take drugs’, ‘to look for a partner’, and ‘to look for sex’ proving to be regarded as least relevant grounds to visit these events.

Spearman correlations between attendance motives are displayed in Table 4, showing significant associations between a wide variety of motives. Especially motives such as ‘to explore my mind’, ‘to escape my daily life’, and ‘to seek excitement’ were significantly correlated with a large number of other motives to attend EDM events.

Comparisons of users versus non-users (i.e., those indicating not to have consumed any substances aside from alcohol in the previous year) (see Figure 2) exhibited differences in motives to attend. Motives such as ‘to dance’, ‘to explore my mind’, ‘to escape my daily life’, ‘to seek excitement’, ‘to drink alcohol’, and ‘to take drugs’ proved to be significantly more important for users compared to their non-using counterparts.

In Table 5, Spearman correlations between motives and frequencies of past-year use are displayed per substance type. The motive ‘to dance’ was most firmly associated with ecstasy/MDMA/Molly but also with cannabis and alkyl nitrites (poppers). ‘To listen to music’ was primarily associated with cannabis and ecstasy/MDMA/Molly and to a lesser extent also with cocaine and amphetamines. The motives ‘to see a particular artist’ and ‘to escape my daily life’ correlated with amphetamines and cannabis. Motives such as ‘to take drugs’, ‘to explore my mind’, and ‘to seek excitement’ were associated with a wide variety of substances (e.g., cannabis, ecstasy/MDMA/Molly, cocaine, ketamine, amphetamines, magic mushrooms, LSD). The motive ‘to open up to my friends’ correlated with alcohol but was negatively associated with cannabis and amphetamines, while ‘to look for sex’ was positively linked with alcohol, alkyl nitrites (poppers), cocaine, cannabis, and other synthetic hallucinogens. The motive ‘to look for a partner’ was associated with other synthetic hallucinogens and cocaine while being negatively related to nitrous oxide.

## 4. Discussion

Previous research on the prevalence of drug use among EDM event attendees generally focused on specific events or particular settings [15,16,17,21,22], while a more exhaustive account of EDM event attendance was still missing. To address this gap, in this study, we surveyed substance use at EDM gatherings in general as well as in specific settings through self-administration. Additionally, we focused on attendance motives and further extended our scope by exploring relationships between these motives and substance use. In contrast to previous records, besides the application of a more embracive approach to examine substance consumption of this specific population, to our knowledge, this is the first study to query motivation for EDM event attendance as well as to interlink both.

First, we sketched a general overview of substance use at EDM settings. Differences in the prevalence of specific substances were shown to depend on the setting, with results indicating ecstasy/MDMA/Molly, other synthetic hallucinogens, ketamine, amphetamines, and alkyl nitrites (poppers) to be the most frequently used substances at festivals, outdoor parties, and raves. In nightclubs, the overall prevalence of ecstasy/MDMA/Molly, ketamine, amphetamines, and alkyl nitrites (poppers) was comparable, while other synthetic hallucinogens were less prevalent. Moreover, cocaine proved to be an often-used substance in nightclubs while being less prominent at festivals, outdoor parties, and raves. Overall, such findings are in line with previous accounts of substance use at public music gatherings (e.g., [8,15,16,17]) as well as at EDM events [9]. Interestingly, in domestic environments, cannabis, magic mushrooms, nitrous oxide, and cocaine were indicated to be the most popular substances although, in general, substance use was less common in such settings except for cannabis and magic mushrooms. At EDM events in pubs/bars, alcohol use proved to be common, while consumption of other substances was shown to be low to negligible.

Overall, aside from alcohol, cannabis was demonstrated to be the most frequently used substance by this particular population. This is somewhat out of tune with previous research signaling an overall dominance of substances such as ecstasy/MDMA/Molly, powder cocaine, LSD, ketamine, and GHB at EDM events (e.g., [8,9,10,17]) and therefore stresses the relevance of a more exhaustive approach regarding this population. Previous work mainly regarded nightclubs and festivals, settings for which our results demonstrated similar trends regarding the prevalence of substances such as ecstasy/MDMA/Molly, cocaine, and ketamine. The overall observed dominance of cannabis on the other hand corresponds to more general findings on substance use which showed that rather than being present at nightlife events, cannabis is primarily used in private settings (e.g., [58]). Thus, our results clearly indicate that setting is key, which should be taken into account when considering drug prevalence at EDM events.

Further, motivation for EDM event attendance was explored, showing that ‘to have fun’ was indicated as the primary reason for attendance, which was closely succeeded by motives relating to music and dance (i.e., ‘to listen to music’, ‘to see a particular artist’, ‘to dance’) as well as to socialization (i.e., ‘my friends are going’). Overall, these results support the idea of a three-dimensional classification of the most principal motives for music event attendance in general, being overall enjoyment, musical enjoyment, and socialization [4,47], and as such, they back conclusions of previous work at music festivals accrediting musical aspects as the most pivotal grounds for attendance (e.g., [46,47,59]). In line with research focusing on cultural festivals [32], a more compelling force of the pull/seeking—rather than the push/escaping dimensions [31,33,38,39,40]—was observed, implying that motivation for EDM event attendance primarily stems from the inherent attractiveness of these events. When taking Getz’s [41,42] motivation paradigm into account, personal motives seemed to overarch physical motives, which was further followed by interpersonal/social ones. Yet, it should be noted that in this study, socialization (i.e., the desire to interact with a group and its members) was regarded in a broader sense. The social dimension with the highest score regarding its motivational properties (i.e., ‘my friends are going’) refers to known-group socialization, i.e., interaction with existing friends/acquaintances. However, external interaction, i.e., socialization with external others who were unacquainted with the visitor prior to the festival (see e.g., [32]) (i.e., ‘to meet new people’), was not considered as an essential instigator by our sample. Furthermore, courtship (i.e., ‘to look for a partner/sex’), which can be part of either of these dimensions, was assessed as most trivial motive for attending. Thus, while EDM event attendees are eager to enjoy themselves and engage with the music, apparently, they mainly want to obtain these experiences in the company of familiar faces.

As could be expected, attendance motives associated to drug use were shown to be more pivotal for illicit substance users than for their non-using counterparts. However, even users regarded drug consumption as a rather trivial instigator for attending. As such, substance use seems to be only a secondary outcome of EDM event attendance rather than it being a motive for visiting such events in the first place. Further distinctions between users and non-users were uncovered, demonstrating substance users to put more emphasis on motives relating to exploration, escapism, excitement, and dance. Such findings thus suggest a stronger drive of users to visit EDM events based on push/escaping motives, pull/seeking forces relating to novelty ([31,32,33,38,39,40]), and physical ones [41,42]. Moreover, this might be linked to the personality trait excitement-seeking, which is a central feature of extraversion and component of the multifaceted impulsivity construct [60]. Psychological reports demonstrate that those scoring high on measures of excitement-seeking are more likely to smoke, use other drugs, and engage in other risky behaviors of clinical and social relevance. Hence, excitement-seeking and risk-taking behavior seems to be relevant to substance use [61]. Moreover, according to behavioral genetic studies, about 50% of the variance in excitement-seeking is heritable [62,63,64]. Thus, from psychological and psychiatric points of view, it would be interesting to explore whether findings such as these could have the potency to confirm a link between substance use and impulsivity-related traits, which might even be related to some genetic variance. Clearly, such argumentation is highly speculative and requires more in-depth inquiry.

Further explorations unveiled correlations between motives for EDM event attendance and specific substances. Cocaine and amphetamine users, for instance, mainly regarded musical motives as key, while ecstasy/MDMA/Molly and cannabis users emphasized motives relating to music as well as dance. Furthermore, attendance motives corresponding to substance use, exploration, and excitement were linked to a rather wide range of substances (e.g., cannabis, ecstasy/MDMA/Molly, cocaine, ketamine, amphetamines, magic mushrooms, LSD). To some extent, our findings correspond to previously demonstrated associations between MDMA use and the desire to experience euphoria, sexiness, self-insight, and sociability/flirtatiousness [65]. Some links could also be made between our results and research focusing on neurological effects of substance use. For example, ecstasy/MDMA/Molly—which was associated with the motive ‘to dance’—has been shown to enhance the release of serotonin and dopamine, as such increasing energy levels while giving the user a sense of euphoria and heightened sensory perception; consequently, it has been commonly used as a ‘rave’ drug [66,67].

Some limitations should be addressed. It should be noted that the rather comprehensive category ‘other synthetic hallucinogens’ was used in the survey, which includes a wide variety of substances of which some have been shown to cause more harm than others [68,69]. In addition, although a variety of recruitment strategies were implemented in order to obtain a representative sample and minimize the risk of non-response bias in this particular study, respondents’ educational levels proved to be rather high. Previous work indeed signaled a general underrepresentation of less highly educated individuals in alcohol and drug population surveys [70]. Another general limitation of research based on alcohol and drug surveys encountered here as well refers to the dependence on retrieved recollections. Using self-assessments of past-year use, a common practice to probe substance use, can never fully circumvent recall bias, which is an issue inherent to survey research in general [71]. Furthermore, due to the sensitive nature of the inquiries, the overall validity of the survey could be questioned. However, self-reports are the primary form of gaining information about substance use [72], and anonymous self-reports were shown to be optimally suited to obtain data regarding sensitive personal issues or issues that, if exposed, could put an individual respondent at risk [72,73,74,75].

## 5. Conclusions

In this study, the prevalence of specific substances at EDM events was shown to depend on the setting. Moreover, our findings support the idea of a three-dimensional classification of the most principal motives for EDM event attendance, being overall enjoyment, musical enjoyment, and socialization, while substance use as such proved to be rather irrelevant as attendance instigator. Finally, some motives could be linked with drug use, which might relate to certain personality traits and neurological effects of these substances. Our novel findings expand previous literature, improving on the understanding of the attendance motives and consumption patterns of EDM event visitors. Such insights have the potential to fuel event management strategies and related prevention and harm reduction efforts, enabling more tailored approaches in these fields of work.

## Figures and Tables

**Figure 1 ijerph-20-01821-f001:**
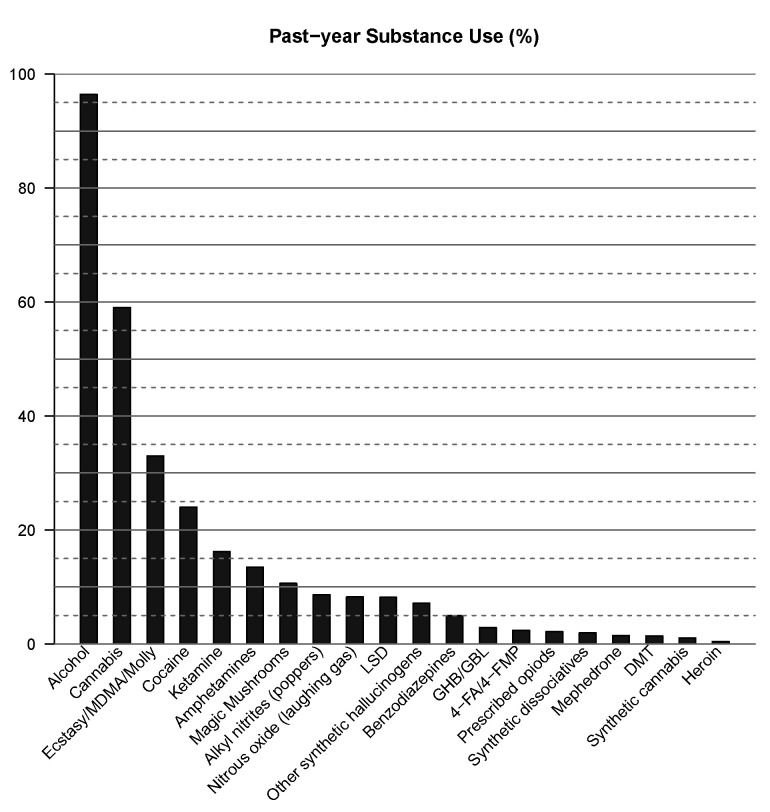
Percentage of participants that consumed the substance in the past year.

**Figure 2 ijerph-20-01821-f002:**
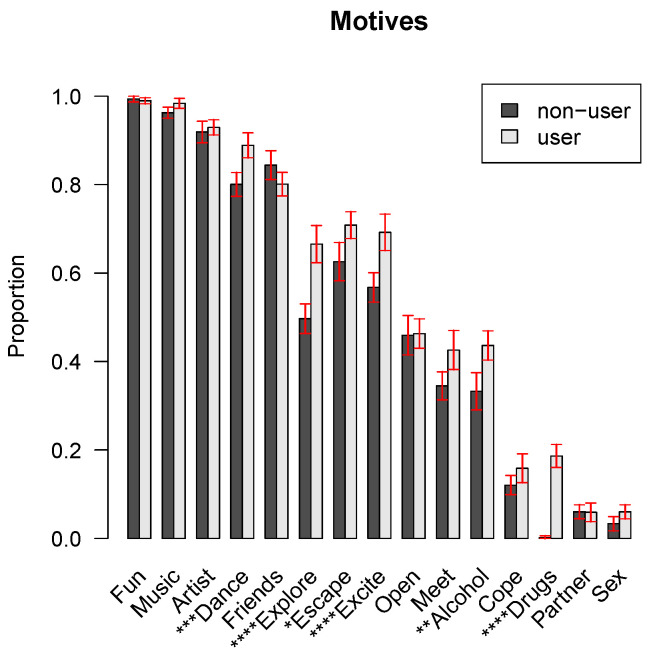
Relevance of EDM attendance motives according to past-year substance use. The bars indicate the proportion of respondents for which the motive was (very) important. A distinction is made between past-year users and non-users, alcohol excluded. The red lines indicate 95% confidence intervals and the magnitude of the Bonferroni-corrected *p*-value comparing non-users with users via logistic regression is denoted by stars (*p* < 0.0001 ****; *p* < 0.001 ***, *p* < 0.01 **, *p* < 0.05 *).

**Table 1 ijerph-20-01821-t001:** Spearman correlations between frequencies of past-year substance use (*p* < 0.0001 ****; *p* < 0.001 ***, *p* < 0.01 **, *p* < 0.05 *).

	Alcohol	Cannabis	Ecstasy/MDMA/Molly	Cocaine	Ketamine	Amphetamines	Magic Mushrooms	Alkyl Nitrites (Poppers)	Nitrous Oxide	LSD
Cannabis	0.04									
Ecstasy/MDMA/Molly	0.14 **	0.24 ****								
Cocaine	0.14 **	0.19 ***	0.47 ****							
Ketamine	−0.08	0.22 ***	0.38 ****	0.24 ***						
Amphetamines	−0.03	0.09	0.30 ****	0.26 ****	0.26 ***					
Magic mushrooms	−0.01	0.28 ****	0.29 ****	0.16 *	0.38 ****	0.08				
Alkyl nitrites (poppers)	−0.01	0.10	0.37 ****	0.23 **	0.18 *	0.18 *	0.18 *			
Nitrous oxide	−0.12	0.20 **	0.27 ***	0.09	0.25 **	0.11	0.07	0.28 **		
LSD	−0.09	0.20 **	0.19 *	0.16 *	0.33 ***	0.06	0.36 ****	0.04	0.12	
Other synthetic hallucinogens	−0.10	0.34 ****	0.31 ***	0.12	0.45 ****	0.18	0.33 ***	0.16	0.13	0.36 ***

**Table 2 ijerph-20-01821-t002:** Location of past-year substance use. The top row displays the number of participants (%) indicating to have used the substance in the past year (in decreasing order from left to right), while the bottom rows reveal the distribution (%) of the location of use.

	Alcohol	Cannabis	Ecstasy/MDMA/Molly	Cocaine	Ketamine	Amphetamines	Magic Mushrooms	Alkyl Nitrites (Poppers)	Nitrous Oxide	LSD	Other Synthetic Hallucinogens
Participants (%) who used the substance in the past year											
	96.43	59.03	33.01	24.01	16.21	13.46	10.63	8.62	8.25	8.18	7.14
Distribution (%) of the location where each substance is most often used											
Nightclub	14.38	1.43	42.73	32.29	24.54	30.90	2.10	23.28	7.21	6.54	15.79
Licensed festival/outdoor party/rave	9.11	9.08	48.64	18.50	33.80	33.15	2.80	29.31	21.62	32.71	48.42
Unlicensed festival/outdoor party/rave	0.47	0.78	3.64	1.25	8.33	7.30	2.10	3.45	10.81	8.41	5.26
Pub/bar	47.76	2.20	0.68	7.84	2.31	1.12	1.40	2.59	2.70	0.00	2.11
Outside/public space	2.20	15.69	0.00	0.31	2.31	1.12	23.08	2.59	9.91	17.76	4.21
Public events	73.92	29.18	95.69	60.19	71.29	73.59	31.48	61.22	52.25	65.42	75.79
House party	9.51	10.64	1.82	16.30	8.80	6.18	4.90	13.79	12.61	2.80	7.37
Own/friend’s home	14.53	58.24	1.82	22.26	18.06	16.85	61.54	24.14	35.14	28.97	13.68
Private events	24.04	68.88	3.64	38.56	26.86	23.03	66.44	37.93	47.75	31.77	21.05
Other	2.04	1.95	0.68	1.25	1.85	3.37	2.10	0.86	0.00	2.80	3.16

**Table 3 ijerph-20-01821-t003:** Percentage of responses for each value of the Likert scale per motive to attend EDM events. Each row sums up to 100%.

	Not Important	Not Very Important	Slightly Important	Very Important (%)
Fun	0.45	0.45	10.86	88.25
Music	0.22	2.16	22.53	75.09
Artist	1.56	5.87	38.14	54.42
Dance	2.16	12.12	40.52	45.20
Friends	2.83	15.54	46.10	35.54
Explore	9.67	29.81	38.74	21.78
Escape	11.23	20.89	46.47	21.41
Excite	9.14	26.10	45.43	19.33
Open	21.41	32.42	38.88	7.29
Meet	11.97	48.33	32.49	7.21
Alcohol	20.37	39.70	34.57	5.35
Cope	53.16	32.34	10.93	3.57
Drugs	61.71	26.25	11.15	0.89
Partner	65.80	28.25	5.50	0.45
Sex	69.22	25.72	4.76	0.30

**Table 4 ijerph-20-01821-t004:** Spearman correlations between attendance motives (*p* < 0.0001 ****; *p* < 0.001 ***, *p* < 0.01 **, *p* < 0.05 *).

	Fun	Music	Artist	Dance	Friends	Explore	Escape	Excite	Open	Meet	Alc	Cope	Drugs	Partner
Music	0.14 ****													
Artist	0.03	0.37 ****												
Dance	0.17 ****	0.16 ****	0.06 *											
Friends	0.10 ***	−0.19 ****	−0.10 ***	0.04										
Explore	0.12 ****	0.28 ****	0.16 ****	0.16 ****	−0.09 **									
Escape	0.12 ****	0.13 ****	0.07 **	0.03	−0.02	0.21 ****								
Excite	0.12 ****	0.16 ****	0.13 ****	0.12 ****	0.01	0.39 ****	0.24 ****							
Open	0.04	−0.03	0.02	0.00	0.24 ****	0.21 ****	0.16 ****	0.19 ****						
Meet	0.06 *	0.07 *	0.04	0.06 *	0.08 **	0.25 ****	0.16 ****	0.23 ****	0.25 ****					
Alc	0.05	−0.13 ****	−0.07 *	−0.02	0.20 ****	−0.02	0.16 ****	0.14 ****	0.17 ****	0.09 ***				
Cope	0.04	0.06 *	0.00	0.04	0.02	0.29 ****	0.31 ****	0.23 ****	0.36 ****	0.20 ****	0.15 ****			
Drugs	0.01	0.15 ****	0.08 **	0.21 ****	−0.06 *	0.24 ****	0.21 ****	0.20 ****	0.02	0.07 *	0.13 ****	0.15 ****		
Partner	0.03	−0.04	0.02	−0.04	0.08 **	0.11 ****	0.21 ****	0.15 ****	0.20 ****	0.31 ****	0.21 ****	0.19 ****	0.08 **	
Sex	0.01	−0.02	0.00	0.03	0.06 *	0.10 ***	0.18 ****	0.21 ****	0.17 ****	0.32 ****	0.27 ****	0.22 ****	0.20 ****	0.61 ****

**Table 5 ijerph-20-01821-t005:** Spearman correlations between attendance motives and frequencies of past-year substance use (*p* < 0.0001 ****; *p* < 0.001 ***, *p* < 0.01 **, *p* < 0.05 *).

	Fun	Music	Artist	Dance	Friends	Explore	Escape	Excite	Open	Meet	Alcohol	Cope	Drugs	Partner	Sex
Alcohol	0.00	−0.03	−0.01	0.04	0.09 ***	−0.03	0.05	0.09 **	−0.01	0.01	0.35 ****	0.04	0.10 ***	0.03	0.13 ****
Cannabis	−0.01	0.16 ****	0.09 **	0.09 **	−0.10 **	0.23 ****	0.10 **	0.10 **	−0.01	0.07 *	−0.09 **	0.10 **	0.48 ****	0.02	0.08 **
Ecstasy/MDMA/Molly	0.11 *	0.13 **	−0.02	0.24 ****	0.00	0.14 **	0.09 *	0.15 ***	0.01	0.04	−0.02	0.02	0.39 ****	0.01	0.07
Cocaine	−0.02	0.11 *	0.04	0.06	−0.01	0.11 *	0.04	0.17 ***	0.04	0.07	0.03	0.03	0.27 ****	0.10 *	0.15 **
Ketamine	0.03	0.06	0.05	0.02	−0.03	0.17 **	0.02	0.03	0.02	0.07	−0.15 *	−0.06	0.12 *	−0.05	−0.07
Amphetamines	0.09	0.13 *	0.16 **	0.04	−0.13 *	0.22 ****	0.16 **	0.18 **	0.02	0.09	−0.04	0.05	0.24 ****	−0.01	0.00
Magic mushrooms	0.00	0.11	0.02	0.04	−0.03	0.13 *	0.00	0.07	0.02	0.07	−0.03	0.14 *	0.24 ****	0.03	0.03
Alkyl nitrites (poppers)	0.06	0.02	−0.03	0.14 *	0.05	0.06	0.11	0.17 *	0.00	0.06	0.05	0.05	0.12	−0.02	0.21 **
Nitrous oxide	−0.03	0.08	−0.03	0.03	−0.05	0.12	0.01	0.09	−0.02	0.12	−0.10	0.04	0.09	−0.16 *	−0.06
LSD	0.14	0.01	−0.09	0.10	−0.07	0.19 **	−0.06	0.09	0.01	0.08	−0.15 *	0.01	0.16 *	−0.01	−0.04
Other synthetic hallucinogens	0.03	0.10	0.00	0.01	−0.04	0.15	0.13	0.14	0.07	0.13	−0.04	0.05	0.28 **	0.20 *	0.18 *

## Data Availability

The data presented in this study are available on request from the corresponding author. The data are not publicly available due to their sensitive nature.

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
