# Peer review of "Substance Use and Attendance Motives of Electronic Dance Music (EDM) Event Attendees: A Survey Study"

_ijerph, 2023, doi:10.3390/ijerph20031821_

Round 1

Reviewer 1 Report

The aim of this study is to explore the EDM event scene and related substance use. 

The authors surveyed 1345 valid survey responses from participants in the EDM scene. They found a ranking of self-reported motivations behind attending EDM activities (fun, friends, coping, etc.), and a ranking of substance use (cannabis, MDMA, etc.). They then did pairwise correlations between the motivations and the substances and found some correlations e.g. between motivation to dance and Molly use, and between alcohol use and sex as a motivation. 

The study is interesting in that it is a first of its kind, but it could use some more methodological rigor. 

It would be good to do correlation matrices within the motivation categories and within the substance use categories as well. More importantly, Please do a factor analysis on the survey items to better understand the statistical structure of your survey.

When comparing users vs non-users, formal statistics should be used, e.g. chi-squared or t-tests. 

Figure 1 “The percentages of past-year substance use among participants are shown in Fig. 1, revealing most frequent use of alcohol, followed by cannabis, ecstasy/MDMA/Molly and cocaine”but alcohol is now shown in figure 1?

Author Response

Please see the attachment (as it includes tables and figures, which could not be added here)

Reviewer 2 Report

This is a good research on such an important subject. The manuscript can be useful in creation of public health policies. However, I think it requires two adjustments:

1. First of all, it is a must to discuss the online collection method. I support it 100%, but such manuscripts should always highlight how important it is to use online surveys in data collection within groups that might be breaking the law, just like illicit substances users. It is needed because such method often lacks validation from a lot of researchers who think it is not adequate to collect data this way when in fact, sometimes it is only way to collect the truth in such popoulations. I recommend two other articles that highlight the importance of anonymous online data collecting within such subjects, please cite them in discussion, both come from Europe and both in rough to handle populations (marihuana cultivators in Belgium and mephedrone users in Poland, both in English):

Decorte "Small Scale Domestic Cannabis Cultivation: An Anonymous Web Survey among 659 Cannabis Cultivators in Belgium" Small Scale Domestic Cannabis Cultivation: An Anonymous Web Survey among 659 Cannabis Cultivators in Belgium - Tom Decorte, 2010 (sagepub.com)

Więckiewicz et al. "Patterns of synthetic cathinones use and their impact on depressive symptoms and parafunctional oral behaviors" pdf-115170-81763 (psychiatriapolska.pl)

2. Unfortunately, you skipped a lot of drugs in your questionnaire, like mephedrone or benzofurans which are still there on illicit market (and imo mephedrone is trending again by the way), this is a limitation that comes to my mind that should also be discussed in limitation section that you could've added more substances. Another thing is, that you don't differ "other synthetic hallucinogens", this should be evaluated in future studies because the amount of people who selected this option is high, and we have some harmless substances like 2-CB and some that could potentially kill and are easy to overdose because of their activity in low amounts, like DOC or DOB. This should also be highlighted in the limitations or in the discussion.

Else than this, the article has it's value and I think that with the upgrades above it is ready to be published. Congratulations.

Author Response

We would like to thank the reviewer for the valuable comments. We have implemented all of them in the revised manuscript. See below for an overview of the changes made, based on the comments of Reviewer 2:

Point 1: First of all, it is a must to discuss the online collection method. I support it 100%, but such manuscripts should always highlight how important it is to use online surveys in data collection within groups that might be breaking the law, just like illicit substances users. It is needed because such method often lacks validation from a lot of researchers who think it is not adequate to collect data this way when in fact, sometimes it is only way to collect the truth in such popoulations. I recommend two other articles that highlight the importance of anonymous online data collecting within such subjects, please cite them in discussion, both come from Europe and both in rough to handle populations (marihuana cultivators in Belgium and mephedrone users in Poland, both in English):

Decorte "Small Scale Domestic Cannabis Cultivation: An Anonymous Web Survey among 659 Cannabis Cultivators in Belgium" Small Scale Domestic Cannabis Cultivation: An Anonymous Web Survey among 659 Cannabis Cultivators in Belgium - Tom Decorte, 2010 (sagepub.com)

Więckiewicz et al. "Patterns of synthetic cathinones use and their impact on depressive symptoms and parafunctional oral behaviors" pdf-115170-81763 (psychiatriapolska.pl)

Actions taken:

In the discussion, when mentioning the limitations of the work, we now added:

“(…) Further, due to the sensitive nature of the inquiries, the overall validity of the survey could be questioned. However, self-reports are the primary form of gaining information about substance use [72] and anonymous self-reports were shown to be optimally suited to obtain data regarding sensitive personal issues or issues that, if exposed, could put an individual respondent at risk [72-75].”

References:

  1. Moore, R.S.; Ames, G.M. Survey confidentiality vs. anonymity: Young men’s self-reported substance use. J. Alcohol Drug Educ. 2002, 47, 32–41.
  2. Richter, L.; Johnson, P.B. Current methods of assessing substance use: A review of strengths, problems, and developments. J. Drug Issues 2001, 31, 809–832.
  3. Decorte, T. Small scale domestic cannabis cultivation: An anonymous web survey among 659 cannabis cultivators in Belgium. Contemp. Drug Probl. 2010, 37, 341–370.
  4. WiÄ™ckiewicz, G.; Smardz, J.; Wieczorek, T.; Rymaszewska, J.; Grychowska, N.; Danel, D.; WiÄ™ckiewicz, M. Patterns of syn-thetic cathinones use and their impact on depressive symptoms and parafunctional oral behaviors. Psychiatr. Pol. 2021, 55, 1101–1119.

Point 2: Unfortunately, you skipped a lot of drugs in your questionnaire, like mephedrone or benzofurans which are still there on illicit market (and imo mephedrone is trending again by the way), this is a limitation that comes to my mind that should also be discussed in limitation section that you could've added more substances. Another thing is, that you don't differ "other synthetic hallucinogens", this should be evaluated in future studies because the amount of people who selected this option is high, and we have some harmless substances like 2-CB and some that could potentially kill and are easy to overdose because of their activity in low amounts, like DOC or DOB. This should also be highlighted in the limitations or in the discussion.

Else than this, the article has it's value and I think that with the upgrades above it is ready to be published. Congratulations.

Actions taken:

In the questionnaire, we did question mephedrone, which is included in Figure 1.

Moreover, the subjects had the option to report on use of other substances, yet we indeed noticed that we did not yet mention this in the manuscript. Therefore, we have now added the following to the methods section:

“(…) including 21 substances/substance classes, alcohol, and the option to indicate other substances”.

Regarding the comment on the category ‘other synthetic hallucinogens’, we now added the following to the discussion:

“It should be noted that the rather comprehensive category ‘other synthetic hallucinogens’ was used in the survey. This however includes a wide variety of substances of which some have been shown to cause more harm than others [68-69].”

References:

  1. Shafi, A.; Berry, A.J.; Sumnall, H.; Wood, D.M.; Tracy, D.K. New psychoactive substances: A review and updates. Ther. Adv. Psychopharmacol. 2020, 10.
  2. Creagh, S.; Warden, D.; Latif, M.A.; Paydar, A. The new classes of synthetic illicit drugs can significantly harm the brain: A neuro imaging perspective with full review of MRI findings. Clin. Radiol. Imaging J. 2018, 2, 000116.

Round 2

Reviewer 2 Report

Well done!